# Benefit or Risk in Patient with Type 1 Diabetes Based on Appropriated Dosage of Dapagliflozin: A Case Report

**DOI:** 10.3390/medicina59050827

**Published:** 2023-04-24

**Authors:** Yan Tian, Weiting Hu, Qun Yan, Bo Feng

**Affiliations:** 1Department of Endocrinology, Shanghai East Hospital, School of Medicine, Tongji University, Shanghai 200120, China; 2Department of Endocrinology, The Second Hospital of Shanxi Medical University, Shanxi Medical University, Taiyuan 030002, China

**Keywords:** T1DM, SGLT2 inhibitors, Dapagliflozin, DKA

## Abstract

*Purpose*: Dapagliflozin has been used extensively in patients with type 2 diabetes mellitus (T2DM). However, due to the potential diabetic ketoacidosis (DKA) risk of dapagliflozin, its use in type 1 diabetes mellitus (T1DM) is limited. Here, we reported an obese patient with T1DM and inadequate glycemic control. We carefully recommended she use dapagliflozin as an insulin adjuvant to achieve better glycemia control and to assess possible benefits and risks. *Methods and Results*: The patient was a 27-year-old female who had underlying T1DM for 17 years with a body weight of 75.0 kg, body mass index (BMI) of 28.2 kg/m^2^, and glycated hemoglobin (HbA1c) 7.7% when admitted. To treat her diabetes, she had used an insulin pump for 15 years (the recent dosage of insulin was 45 IU/d) and oral metformin for 3 years (0.5 g qid). In order to decrease body weight and achieve better glycemic control, dapagliflozin (FORXIGA, AstraZeneca, Indiana) was administered as an insulin adjuvant. The patient presented sever DKA with a euglycemia (euDKA) after two days of the administration of dapagliflozin at a dose of 10 mg/d. euDKA occurred again after the administration of dapagliflozin at a dose of 3.3 mg/d. However, after using a smaller dose of dapagliflozin (1.5 mg/d), this patient achieved better glycemia control, with a significant reduction in daily insulin dosage and gradual weight loss, without significant hypoglycemia or DKA occurring. At the sixth month of the administration of dapagliflozin, the HbA1c was 6.2% for the patient, her daily insulin dosage was 22.5 IU, and her body weight was 60.2 kg. *Conclusions*: The appropriate dose of dapagliflozin is critical for a patient with T1DM patient therapy in order to find a correct balance between the benefits and risks.

## 1. Introduction

T1DM occurs mainly in adolescents; insulin replacement therapy is the mainstay of therapy for T1DM patients [1]. Despite the fact that great progress has been made in recent years in terms of insulin delivery and glucose continuous monitoring systems, glycemic control in T1DM patients is often suboptimal, with less than a third of this population achieving optimal glycemic control [2]. Sodium glucose cotransporter 2 (SGLT2) inhibitors are a new type of hypoglycemic drugs that inhibit renal glucose reabsorption in the proximal convoluted tubule, leading to increased urinary glucose excretion [3]. Additional benefits include weight loss and protection against cardiovascular and kidney diseases. Dapagliflozin, as a SGLT2 inhibitor, has been approved by the US Food and Drug Administration as a new class of glucose-lowering agents for the treatment of T2DM, either as monotherapy or as an add-on treatment [4]. Dapagliflozin has also shown to improve glycemic control and reduce the total daily insulin dose in patients with T1DM. However, the favorable efficacy profile of dapagliflozin needs to be balanced against possible side effects, in particular, diabetic ketoacidosis (DKA). DKA is a serious and potentially life-threatening acute complication of diabetes which is characterized by hyperhemoketosis and dehydration [5]. T1DM patients are more prone to DKA due to absolute insulin deficiency. In addition, SGLT2-inhibitors-induced DKA usually presents euglycemia (plasma glucose is <11 mmol/L), making it challenging to identify and to find therapy for in clinical practice [6]. Since several randomized controlled clinical trials assessing the efficacy and safety of dapagliflozin as an adjuvant therapy to insulin in individuals with T1DM have been published [7,8], whether there is more of a benefit or risk to optimize this compound in this indication has been disputed.

Here, we showed that DKA occurred in a patient with T1DM after 10 mg or 3.3 mg of dapagliflozin usage, but the small dose of dapagliflozin (1.5 mg/d) resulted in better glycemic control and a gradual decrease in body weight and daily insulin dosage without the occurrence of severe hypoglycemic events and DKA. We aim to provide more evidence in favor of an improved benefit/risk ratio using dapagliflozin in T1DM patients.

## 2. Case Report

A 27-year-old female, with underlying T1DM for 17 years, was recently admitted to the hospital in order to achieve better glycemic control. Vital signs at presentation were a temperature of 36.1 °C, a pulse rate of 100 beats/min, a respiratory rate of 28 breaths/min, and blood pressure of 120/85 mm Hg. Past medical history included T1DM managed with an insulin pump 15 years. In recent years, the dosage of insulin was 45 IU/d, and metformin 0.5 g (qid) was added to the therapeutic regimen 3 years ago due to weight gain. On admission, characteristics were noted as follows: her body weight was 75.0 kg; BMI was 28.2 kg/m^2^; and waist circumference was 105.5 cm. Clinical laboratory analysis found that she was positive for urine ketones, her fasting blood glucose was 10.5 mmol/L, and her glycated hemoglobin (HbA1c) was 7.7%. Her was C-peptide <0.02 ng/mL, insulin antibodies (IAA) were 4.96 RU/mL, islet cell antibodies (ICA) were 6.09 IU/mL, and glutamic acid decarboxylase antibodies (GAD) were 18.44 IU/mL. An abdominal ultrasound indicated that she had a fatty liver. An electrocardiograph (ECG) and auscultation of the lungs showed no significant findings; her liver function, renal function, and thyroid function had no obvious abnormalities. The patient had no smoking and alcohol intake history, but she did have a history of fracture surgery due to a traffic accident and a family history of diabetes.

In order to decrease body weight and achieve better glycemic control, 10 mg/d of dapagliflozin (FORXIGA, AstraZeneca, Indiana) was being administered as part of a new therapeutic schedule. After two days of dapagliflozin usage, she presented with asthenia, dyspnea, nausea, vomiting, and a poor oral intake. Arterial blood gases showed a picture of severe metabolic acidosis with an elevated anion gap (pH 7.285, CO_2_ 31.80 mmHg, bicarbonate 15.00 mmol/L, base excess −11.06 mmol/L), but her presented euglycemia (<11 mmol/L) and serum lactate levels were normal (0.80 mmol/L). She was treated with a balanced saline solution and 5% glucose liquid with the addition of an insulin infusion (total 3000 mL/d) through intravenous rehydration for a successive five days. Serial blood gas analyses showed a gradual resolution of her ketoacidosis with a normalized anion gap. She was re-treated with a small dosage of dapagliflozin (3.3 mg/d) when her condition stabilized. However, after two days, she re-presented with asthenia, dyspnea, and nausea. Arterial blood gases showed that the pH was 7.269. Dapagliflozin was permanently discontinued. The clinical features of DKA induced by a different dose of dapagliflozin are shown in Table 1. The patient was discharged after 4 weeks of hospitalization; her therapeutic schedule was 29.2 IU/d of insulin and 0.5 g of metformin (qid).

After two weeks, the patient began self-administration with a lower dose of dapagliflozin (1.5 mg/d) to acquire satisfactory glycemic control; this gradually decreased her body weight and insulin dosage. In this period, her dietary structure was not adjusted subjectively, and the amount of exercise she did was not altered.

### 2.1. Changes in Body Weight and Insulin Dosage

The patient’s body weight at hospitalization was 75.00 kg and her BMI was 28.23 kg/m^2^, meaning she was defined as overweight. As shown in Figure 1a, severe DKA occurred on the 7th day after 2 days of the administration of 10 mg/d of dapagliflozin, where a rapid reduction in body weight can be seen. The second DKA also caused rapid weight loss. On the 45th day, her body weight gradually decreased after the administration of 1.5 mg/d of dapagliflozin was applied. Similarly, as shown in Figure 1b, the dosage of insulin significantly decreased after the addition of dapagliflozin to the treatment strategy. After discharge, the patient increased her daily insulin because of poor glycemic control (day 28–45). After being treated with 1.5 mg/d of dapagliflozin, her insulin dosage gradually decreased. By the 180th day, the patient weighed 60.2 kg and her daily insulin dosage was 22.5 IU.

### 2.2. Safety and Efficacy

As shown in Table 2, the patient’s BMI, blood glucose, urine ketones, waist circumference, and HbA1c had significantly decreased, which indicated satisfactory glycemic control after the administration of 1.5 mg/d of dapagliflozin. In addition, no severe hypoglycemia or DKA occurred during the time when 1.5 mg/d of dapagliflozin was being administered, but urinary ketone overload occurred twice, and this was detected by urine ketone self-monitoring. Particularly, we noticed that 1.5 mg/d of dapagliflozin resulted in an increase in urinary glucose, indicating the effectiveness of a low dose of dapagliflozin for this patient. Furthermore, continuous glucose monitoring (CGM, Abbott Laboratories, Shanghai, China) was used to investigate the low dose of dapagliflozin for the stability of glycemic control. As show in the average daily blood glucose graph (Figure 2), the patient’s average blood glucose was 9.4 mmol/L, and her estimated HbA1c was 7.5% during hospitalization, while when 1.5 mg of dapagliflozin was administered, the patient achieved a stabilized and satisfying glycemic control, her average blood glucose was 7.2 mmol/L, and her estimated HbA1c was 6.2%.

### 2.3. 16S rRNA Sequencing

16S rRNA sequencing was performed to evaluate nutrition metabolism 3 days before the patient’s hospitalization and during the first DKA using stool specimens by Shanghai Biozeron Company. Due to the reason that the three major nutrients of the nutrition metabolism are critical in diabetes and ketoacidosis pathology, we mainly analyzed changes in the proportion of protein, fat, and carbohydrates. To the sequencing data we performed a function analysis to predict the microbiome-related content of the nutrient metabolism using Picrust software. The results showed that the protein ratio was 6.41% before DKA, which was below the normal range, and the ratio of fat and carbohydrates was not significantly abnormal. When DKA occurred, the balance between carbohydrates and fat shifted to fat metabolism (Figure 3).

## 3. Discussion

We reported an obese patient with T1DM and inadequate glycemic control. We carefully recommended she use dapagliflozin as an insulin adjuvant to achieve better glycemia control and to receive possible benefits. The patient presented euDKA after the administration of dapagliflozin at dose of 10 mg/d and 3.3 mg/d. However, after using a smaller dose of 1.5 mg/d, this patient achieved better glycemia control, with a significant reduction in daily insulin dosage, and gradual weight loss, and without the occurrence of significant hypoglycemia or DKA.

SGLT2 inhibitors are the newest class of antihyperglycemic medications. The side effect of SGLT2 inhibitors is that there is a certain risk of DKA, which makes the use of SGLT2 inhibitors in patients with T1DM controversial. However, as SGLT2 inhibitors can reduce the average glycemia and insulin dosage while promoting weight loss, the off-label use of SGLT2 inhibitors in the setting of T1DM patients is increasing. In DEPICT-1 and DEPICT-2 studies of the efficacy and safety of dapagliflozin, patients with T1DM treated with daily doses of 5 mg or 10 mg of dapagliflozin saw improvements in their glycemic control and weight loss, although there was also an incidence rate of DKA of 4.0%, though most of these events were of mild or moderate severity, and they were all resolved with treatment [9]. In a study of patients with BMI ≥ 27 kg/m^2^, the incidence of DKA was only 1.7% [10]. Based on those positive results, the administration of 5 mg of dapagliflozin received marketing approval in Europe in March 2019 as an adjuvant therapy to insulin in people with T1DM with a BMI ≥ 27 kg/m^2^ (https://www.europeanpharmaceuticalreview.com/news/73381/ema-forxiga-type-1-diabetes/ (6 March 2018)). In another study on the efficacy and safety of SGLT2 inhibitors in patients of T1DM, it was found that 2.5 mg of empagliflozin can also significantly reduce HbA1C and body weight, but the DKA rate was lower than that of that at administrations of 10 mg and 25 mg, and the DKA rate was similar to that of the placebo group [11]. This reminds us that if the treatment of dapagliflozin is started from a minimum dose, a certain percentage of DKA for those of T1DM patients in a DEPICT study may be avoided. Our current case seems to confirm this.

DKA is one of the most serious and potentially life-threatening complications of diabetes. The incidence of DKA is between 4.6 and 8.0 cases per 1000 persons/years among diabetic patients [12]. DKA is traditionally defined by the symptoms of hyperglycemia (>13.9 mmol/L), increased plasma ketones, and anion-gap acidosis [13]. EuDKA, defined as DKA with euglycemia, is classically considered rare, but this is perhaps a result of under recognition. Reports of SGLT2-inhibitor-associated euDKA have been showed in patients with both T1DM and T2DM [14]. The absence of significant hyperglycemia in many of these patients may delay recognition of the emergent nature of the problem. The mechanism of euDKA has not yet been fully elucidated. SGLT2 inhibitors could induce a sustained urinary glucose loss of 40–80 g/day under conditions of normal glycemia [15]. One of the causes of the euDKA may be due to a lower renal threshold for glucose and a loss of large amounts of glucose in the urine in the presence of an increased rate of gluconeogenesis and free fatty acid release. In addition, a sharp decrease in insulin dosage may also be an important reason for euDKA. Insulin lowers ketone levels by inhibiting lipolysis and hepatic ketogenesis as well as increasing the oxidation of ketones in the peripheral tissue. The reduced insulin doses at a time of heightened insulin resistance may have triggered the balance toward ketosis, resulting in euDKA. In this case, the daily insulin dosage decreased by nearly 50% when DKA events occurred; the occurrence of DKA is also related to a decrease in insulin. Additionally, it was reported that SGLT2 inhibitors could reduce the renal tubular clearance of ketone bodies [16]. This condition may increase the blood ketone body concentration and increase the risk of ketoacidosis. The results of 16S rRNA sequencing suggest that the patient was severely deficient in glucose when DKA occurred, and their energy supply was converted from glucose to fat. On the other hand, at the time of DKA, the patient had a very low carbohydrate level, indicating that a relatively high dose of dapagliflozin led to metabolic disturbances. However, more evidence needs to be collected to support this finding. Furthermore, another important benefit observed was weight loss. There was a significant decrease in weight (from 75.0 kg to 60.2 kg) and waist circumference (from 105.5 cm to 88.0 cm) in this patient. Weight loss with SGLT2 inhibitor therapy has been consistently observed in several studies in T2DM and T1DM; the benefits of these inhibitors has also been noted in a few studies with obese subjects without diabetes [17]. However, the magnitude of weight loss is modest both in patients with diabetes and those with obesity without diabetes. For approved SGLT2 inhibitors, there was, on average, some 1.5–3 kg of weight loss (placebo adjusted) [9,17,18], and these effects are dose dependent [19]. The mechanisms of attenuated weight loss due to SGLT2 inhibitors are not well known. The direct reason is that SGLT2 inhibitors directly cause body weight loss via glucose excretion (calorie loss) in the kidneys. The inhibition of SGLT2 acts in a glucose-dependent manner and can result in the elimination of about 60–100 g of glucose per day in the urine. On the other hand, the SGLT2 inhibitor enhanced lipolysis and shifted substrate usage from carbohydrates to lipids [20]. In addition, Akihiro Yoshida et al. [21] found that SGLT2 inhibitors can improve the antilipolytic effect and lead to greater weight loss for those who have higher levels of adipo-insulin resistance. From this perspective, our present patient’s significant weight loss may be related not only to the effect of SGT2i on glucose excretion and a reduced dosage of insulin use, but also to a significant improvement in adipose insulin resistance.

In clinical practice, due to individual differences, clinicians must adjust the dose of dapagliflozin by closely monitoring the patients’ glycemia and serum or urinary ketone bodies to reduce the risk of DKA. In addition, the speed of DKA occurring can also be used as an indicator to determine what the appropriate dosage is. In our case, the patient developed severe DKA only after two days of dapagliflozin administration, indicating that a 10 mg or 3.3 mg dosage is relatively large, and a lower dose may result in safer clinical practice.

## 4. Conclusions

We reported that a T1DM patient presented DKA as a complication of dapagliflozin (10 mg/d and 3.3 mg/d), while a 1.5 mg/d dose of dapagliflozin resulted in better glycemia control, with a significant reduction in daily insulin dosage and gradual weight loss, without the occurrence of significant hypoglycemia or DKA. The appropriate dose of SGLT2 inhibitors is critical for patients with T1DM in terms of striking the correct balance for the patients’ therapy, considering the benefits and risks.

## Figures and Tables

**Figure 1 medicina-59-00827-f001:**
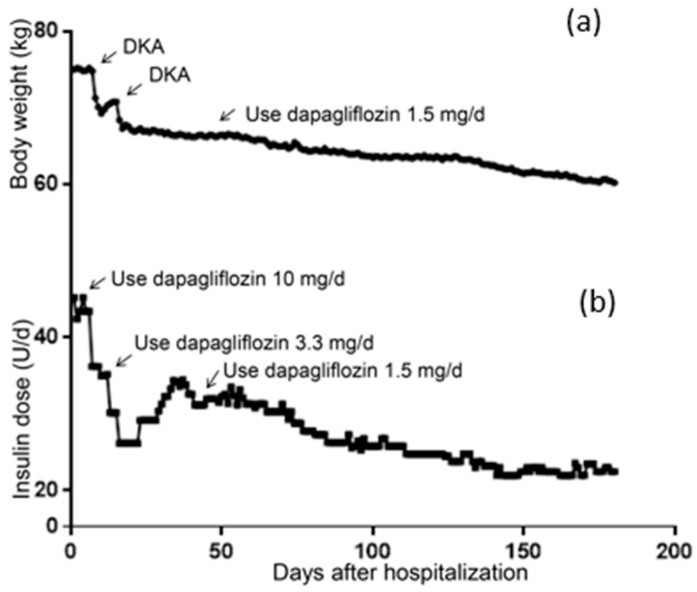
Changes of body weight and daily insulin dosage. (**a**) Change of body weight within 6 months. (**b**) Change of daily insulin dosage within 6 months.

**Figure 2 medicina-59-00827-f002:**
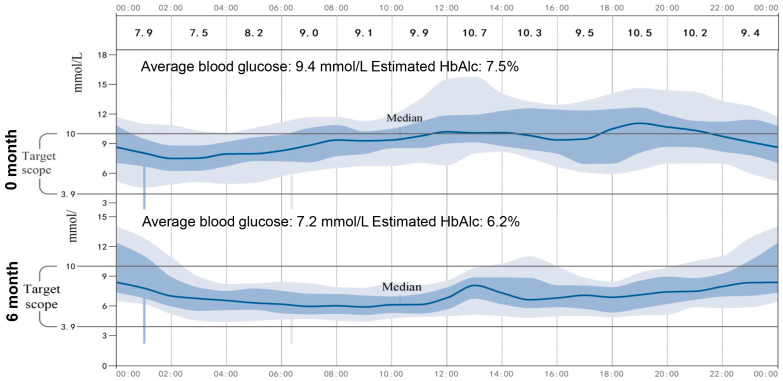
Average daily blood glucose graph. The CGM continued for 14 days, the average daily blood glucose changes graph at day 0 and after 6 months was showed.

**Figure 3 medicina-59-00827-f003:**
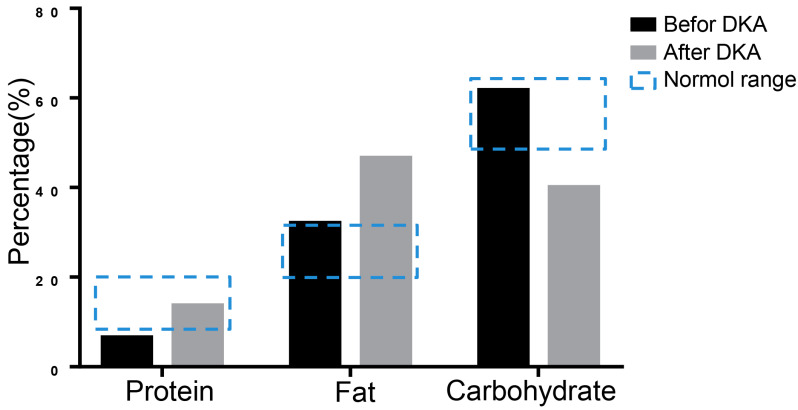
The results of 16S rRNA sequencing. The metabolism was analyzed using 16S rRNA sequencing of stool specimen. The predicted proportions of protein, fat, and carbohydrate were showed.

**Table 1 medicina-59-00827-t001:** Comparison of different dosage of dapagliflozin induced DKA.

The Dosage	10 mg	3.3 mg	1.5 mg
DKA occurring?	Yes	Yes	No
Days of DKA occurring after drug use	2	2	
Insulin dosage (IU/d)	36.2	30.2	22.5
Fasting Blood glucose (mmol/L)	6.0	6.3	
BMI (kg/m^2^)	26.84	26.65	22.7
Arterial blood PH	7.285	7.269	
Duration of DKA (Day)	5	4	

**Table 2 medicina-59-00827-t002:** Safety and efficacy of dapagliflozin on glycemic control.

Parameter	0 Month	6 Month
BMI (kg/m^2^)	28.2	22.7
Waistline (cm)	105.5	88.0
Fasting blood glucose (mmol/L)	7.6	4.9
Postprandial blood glucose (mmol/L)	14.2	9.4
Urine glucose	+	+++
Urine ketones	+	+
HbA1c (%)	7.7	6.3
C-peptide (ng/mL)	<0.02	<0.02

## Data Availability

Data is unavailable due to privacy.

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
