# Peer review of "Benefit or Risk in Patient with Type 1 Diabetes Based on Appropriated Dosage of Dapagliflozin: A Case Report"

_medicina, 2023, doi:10.3390/medicina59050827_

Round 1

Reviewer 1 Report

The study interesting but following changes are required before publication.

1.       There should be the uniformity throughout the manuscript. 

2.       Manuscript must be proof read and grammatic and basic English language errors should be resolved

3.       Abstract must be rewritten. Author must focus on results along with objectives of the research. In the abstract section author focused more on objectives and background of the study instead of results. Results are discussed under the section of methods, there should be a heading of results.

4.       When abbreviations for type 1 diabetes (TID) and type 2 diabetes (T2D) are mentioned in abstract then there is no need to mentioned again both full form and abbreviation in introduction or anywhere in whole manuscript. Only given abbreviation must be mentioned throughout the manuscript.

5.       First part of conclusion is too much confusing. Which dose of dapagliflozin is recommended.

6.       In bibliography list reference number 14 is too old. Updated reference must be added.

Author Response

Thank you for your letter and for the reviewers’ comments concerning our manuscript entitled “Benefit or risk of dapagliflozin in type 1 diabetes patient: A case report” (ID: medicina-2284702). Those comments are valuable and helpful for revising and improving our paper. We have carefully made correction according comments, and hope it meet with approval. Revised portions are highlighted with yellow in the paper. The main corrections in the paper and the responds to the reviewer’s comments are as flowing:

Reply to Reviewer 1

Q1: There should be the uniformity throughout the manuscript.

A1: Thank you for your comments. We have reorganized the article to make it more consistent now.

Q2: Manuscript must be proof read and grammatic and basic English language errors should be resolved.

A2: Thank you for your comments. We are very sorry for our incorrect written. The manuscript has been edited and polished by recommend language editing service now.

Q3: Abstract must be rewritten. Author must focus on results along with objectives of the research. In the abstract section author focused more on objectives and background of the study instead of results. Results are discussed under the section of methods, there should be a heading of results.

A3:  Thank you for your kind suggestions. We have rewritten the abstract now, which is more organized and more focused now.

Q4: When abbreviations for type 1 diabetes (TID) and type 2 diabetes (T2D) are mentioned in abstract then there is no need to mentioned again both full form and abbreviation in introduction or anywhere in whole manuscript. Only given abbreviation must be mentioned throughout the manuscript.

A4: I’m sorry for this mistake, it has been modified.

Q5: First part of conclusion is too much confusing. Which dose of dapagliflozin is recommended.

A5: Apologies, we have not demonstrated this clearly enough in the Discussion section; the first paragraph of the Discussion was indeed a bit confusing. We have now revised this. In the first paragraph, we have written a summary, while the second paragraph explains that selecting the appropriate dose of dapagliflozin is necessary in order to balance out the risks and benefits. We hope that is now clearer to understand.

Q6: In bibliography list reference number 14 is too old. Updated reference must be added.

A6: Reference 14 has been updated now.

Reviewer 2 Report

This is an interesting clinical case assessing the efficacy and safety of dapagliflozin in a T1D patient on insulin and metforin. The authors reported a dose dependent occurrence of DKA on 10 and 3.3 mg of dapagliflozin, while the drug was efficient and well-tollarated on lower dose. In total, the study is well-written and depicts a clinical meaningful aspect by presenting some novel data (16S rRNA sequencing results). However, a major issue should be clarified:

-Figure 1b depicts that the initiation of 10 and 3.3 mg of dapagliflozin was associated with a concomitant sharp decrease in insulin dose, but this wasn't the case on dapagliflozin 1-2mg. Given that this decrease in insulin dose can be a major contribution to the DKA events, it is rational to assume that the former, and not dapagliflozin dose per se, can result in DKA. Please include a relevant discussion in the "Discussion"

Author Response

Thank you for your letter and for the reviewers’ comments concerning our manuscript entitled “Benefit or risk of dapagliflozin in type 1 diabetes patient: A case report” (ID: medicina-2284702). Those comments are valuable and helpful for revising and improving our paper. We have carefully made correction according comments, and hope it meet with approval. Revised portions are highlighted with yellow in the paper. The main corrections in the paper and the responds to the reviewer’s comments are as flowing:

Reply to Reviewer2

Q1:Figure 1b depicts that the initiation of 10 and 3.3 mg of dapagliflozin was associated with a concomitant sharp decrease in insulin dose, but this wasn't the case on dapagliflozin 1-2mg. Given that this decrease in insulin dose can be a major contribution to the DKA events, it is rational to assume that the former, and not dapagliflozin dose per se, can result in DKA. Please include a relevant discussion in the "Discussion"

A1: Thank you for your constructive suggestions. In the present case, the daily insulin dosage decreased by nearly 50% when DKA events occurred. A sharp decrease in insulin dosage may also be an important reason for the occurrence of DKA in the present case. Insulin lowers ketone levels by inhibiting lipolysis and hepatic ketogenesis, as well as increasing the oxidation of ketones in the peripheral tissue. The reduced insulin doses at a time of heightened insulin resistance may have triggered the balance toward ketosis, resulting in DKA. In this case, the occurrence of DKA was also related to a decrease in insulin. We have revised the Discussion section; please see the third paragraph of the Discussion for the relevant revisions.

Reviewer 3 Report

The authors might explain how the patient took the actual dose of 1-2 mg/day of dapagliflozin and why the patient self administrated this drug.

The authors might also comment other potential causes for weight loss since the patient experimented an important weight loss, amount that is not sustained by the SGLT2i trials.

Author Response

Thank you for your letter and for the reviewers’ comments concerning our manuscript entitled “Benefit or risk of dapagliflozin in type 1 diabetes patient: A case report” (ID: medicina-2284702). Those comments are valuable and helpful for revising and improving our paper. We have carefully made correction according comments, and hope it meet with approval. Revised portions are highlighted with yellow in the paper. The main corrections in the paper and the responds to the reviewer’s comments are as flowing:

Reply to Reviewer 3

Q1: The authors might explain how the patient took the actual dose of 1-2 mg/day of dapagliflozin and why the patient self administrated this drug.

A1: The patient used a grinding machine to control the dosage of dapagliflozin. The device crushes the pills. For example, when the dose of dapagliflozin was 10 mg, this weighed 1 gram, and 1.5 mg should weigh 0.15 g. On average, when one tablet was crushed and weighed, this resulted in seven portions.

The patient is a medical graduate student. We fully explained the advantages and disadvantages of the drug to her. After DKA occurred twice, we decided to discontinue the use of dapagliflozin, and we again informed the patient of the risks. Due to the patient’s strong willingness to lose weight, she attempted to self-administer this drug with carefully monitoring her urine ketones every day.

Q2: The authors might also comment other potential causes for weight loss since the patient experimented an important weight loss, amount that is not sustained by the SGLT2i trials.

A2:  Thank you for your constructive comments. The magnitude of weight loss for SGT2i is modest both in diabetes and in obesity without diabetes. For approved SGLT2 inhibitors there is on average some 1.5–3 kg weight loss(placebo-adjusted) [doi: 10.1111/dom.12670; doi: 10.1111/dom.14248; doi: 10.1002/oby.20663], and these effects are dose-dependent [doi: 10.1002/oby.22066]. The mechanisms of the attenuated weight loss due to SGLT2is are not well known. The direct reason is that SGLT2 inhibitors directly cause body weight loss via glucose excretion (calorie loss) in the kidneys. Inhibition of SGLT2 acts in a glucose-dependent manner and can result in the elimination of about 60–100 g of glucose per day in the urine. On the other hand, SGLT2 inhibitor enhanced lipolysis and shifted substrate usage from carbohydrates to lipids. In addition, Akihiro Yoshida et al. found that SGLT2 inhibitors can improve antilipolytic effect and lead to a greater weight loss for those whose have higher levels of adipo-insulin-resistance [doi: 10.1210/jc.2018-02254] . From this perspective, our present patient's significant weight loss (from 75.0kg to 60.2kg) may be related not only to the effect of SGT2i on glucose excretion and reduced dosage of insulin use, but also to a significant improvement in adipose insulin resistance. We have revised the discussion accordingly. Please see the fifth paragraph of Discussion.